# Development and Evaluation of the *Canteen Connect* Online Health Community: Using a Participatory Design Approach in Meeting the Needs of Young People Impacted by Cancer

**DOI:** 10.3390/cancers14010050

**Published:** 2021-12-23

**Authors:** Jennifer Cohen, Pandora Patterson, Melissa Noke, Kristina Clarke, Olga Husson

**Affiliations:** 1Canteen, Sydney, NSW 2042, Australia; jennifer.cohen@canteen.org.au (J.C.); melnoke@gmail.com (M.N.); kristina.clarke@canteen.org.au (K.C.); 2School of Women’s and Children’s Health, University of NSW, Sydney, NSW 2031, Australia; 3Faculty of Medicine and Health, The University of Sydney, Sydney, NSW 2006, Australia; 4Medical Oncology & Psychosocial Research and Epidemiology Department, Netherlands Cancer Institute, 1105 AZ Amsterdam, The Netherlands; o.husson@nki.nl

**Keywords:** AYAs, online health community, cancer, participatory design, peer support, psychosocial well being

## Abstract

**Simple Summary:**

Adolescent and young adults (AYAs) impacted by their own or familial cancer require information and peer support throughout the cancer journey to help with their feelings of isolation. AYAs impacted by cancer need safe, secure, and accessible ways to connect with their peers and access information, peer, and professional support. Online Health Communities provide social networks, support, and health-related content to people united by a shared health experience. Canteen Connect (CC) was developed using a participatory design (PD) process, covering a needs assessment, idea generation, and implementation evaluation. The evaluation showed CC was appropriate for connecting with other AYAs. Most AYAs reported satisfaction with CC and a positive impact on their feelings of sadness, worry, and/or anxiety. By using a PD approach, CC fills an important service provision gap in providing an acceptable and appropriate online health community for AYAs impacted by cancer, with initial promising psychological outcomes.

**Abstract:**

Adolescent and young adults (AYAs) impacted by their own or familial cancer require information and peer support throughout the cancer journey to ameliorate feelings of isolation. Online Health Communities (OHC) provide social networks, support, and health-related content to people united by a shared health experience. Using a participatory design (PD) process, Canteen developed Canteen Connect (CC), an OHC for AYAs impacted by cancer. This manuscript outlines the process used to develop CC: (1) A mixed-methods implementation evaluation of Version I of CC (CCv.1); (2) Qualitative workshops utilizing strengths-based approaches of PD and appreciative inquiry to inform the development of CC Version 2 (CCv.2); quantitative implementation evaluation to assess the appropriateness, acceptability, and effectiveness of CCv.2. Through several iterations designed and tested in collaboration with AYAs, CCv.2 had improvements in the user experience, such as the ability to send a private message to other users and the site becoming mobile responsive. Results from the evaluation showed CCv.2 was appropriate for connecting with other AYAs. Most AYAs reported satisfaction with CCv.2 and a positive impact on their feelings of sadness, worry, and/or anxiety. CCv.2 fills an important service provision gap in providing an appropriate and acceptable OHC for AYAs impacted by cancer, with initial promising psychological outcomes.

## 1. Introduction

Adolescence and young adulthood are times of significant change to family, peer, and intimate relationships; identity; independence and living situation; and work and study goals [1,2]. The disruption resulting from a familial (either parent or sibling) or personal cancer diagnosis or bereavement can lead to considerable psychosocial impacts on adolescent and young adults (AYAs) during this vulnerable developmental period [3,4,5,6]. Between 47–60% of Australian AYAs affected by a personal cancer diagnosis, or familial cancer diagnosis or bereavement, have reported experiencing high or very high psychological distress [3,4,5,7]. This is a greater proportion than the 9–17% reported in general Australian samples and places AYAs at risk for developing mental health disorders, such as anxiety, depression, or post-traumatic stress disorder [8,9].

AYAs impacted by their own or a familial cancer diagnosis report unmet needs for information, family communication, social support, help dealing with difficult emotions, and psychological support, both during and after the cancer experience [10,11]. Addressing young people’s unmet needs can act as a protective factor against poor psychosocial outcomes and promote well-being [10,12]. For example, receiving social support reduces isolation and promotes psychosocial quality of life for AYA cancer survivors [13]. There is a need for comprehensive services that include information, peer support, and counseling to meet the unmet psychosocial needs of this vulnerable population of AYAs [14].

Digital technology plays a significant role in young people’s lives [6]. Utilizing digital health technology to support young people impacted by cancer is considered an integral part of supportive care [6] and shows promise for improving outcomes across multiple health domains [15]. There are multiple supportive care digital health interventions developed for AYAs with and beyond cancer, and these have shown good acceptability [16]. Unfortunately, most of these supportive care digital interventions have not been implemented at scale [16]. Online Health Communities (OHC) are a form of digital health technology that provide online social networks, support, and health-related content to people united by a shared health experience [16]. OHC allow people to seek health information and share illness experiences online with other people in a similar situation [17]. OHC enable marginalized populations, such as those in rural communities or young people impacted by cancer, to connect and foster a unique information-sharing environment [18,19]. OHC allow individuals to identify with others through shared experiences and openly express their views in a safe space [19]. For older, adult cancer patients, engagement in OHC has been associated with psychological benefits such as emotional relief, reduced stress, increased sense of control, and post-traumatic growth [16,20].

Despite technology being an essential part of young people’s lives [21,22], using digital technology in the form of OHC for young people impacted by cancer remains in its infancy, with few secure environments available that enable AYAs to connect to dedicated peer and professional support [23]. The OHC that do exist for AYAs diagnosed with cancer have shown promise, leading to improvements in AYAs’ ability to connect with others through peer support [23,24]; self-efficacy [25]; skills to manage their cancer experience [26,27]; and quality of life [28]. Almost a decade ago, an OHC for young people in Australia impacted by their own or a familial cancer diagnosis was recognized as necessary to provide much-needed access to online support, such as counseling, cancer information, and peer support [14]. Although there are several OHC for young people with cancer, there were no OHC available at that time to address the unmet support needs of AYAs impacted by a familial cancer diagnosis or bereavement.

### Development of Canteen Connect

The development and evaluation of Canteen Connect (CC), an OHC for young people impacted by their own or a familial cancer diagnosis, occurred over two phases (see Figure 1). Phase One covered the delivery and implementation evaluation of the first version of CC (CCv.1). Implementation evaluation is an essential component in developing healthcare solutions to ensure the program or service is effective, sustainable, and integrated into routine care [29]. Using the implementation evaluation framework of Proctor et al. 2011 [30], a mixed-methods implementation evaluation was undertaken to assess users’ views of the acceptability and appropriateness of CCv.1. Phase Two covered the needs assessment and idea generation, development, testing and delivery, and implementation evaluation of the second version of CC (CCv.2) utilizing participatory design (PD). This paper presents the implementation evaluation of CCv.1 (Study I); the PD phases of needs assessment and idea generation (Study II); and the implementation evaluation of CCv.2 (Study III).

## 2. Study I: Mixed-Methods Implementation Evaluation: Canteen Connect Version 1

In 2014, Canteen launched an OHC for young people impacted by personal or familial cancer [14]. The OHC included discussion forums for users to interact with other young people impacted by cancer, blog posts, stories from other young people, as well as access to an online counseling service. The OHC was moderated by health professionals experienced in working with young people impacted by cancer and trained to encourage active participation, monitor user activity, and report user safety concerns. Canteen’s OHC aimed to provide a safe space for young people to explore and express their feelings, build their coping strategies, and enjoy time out from the pressures of daily life [14]. In 2018, after five years of OHC delivery and refinement, an implementation evaluation was conducted on CCv.1. Drawing from the implementation evaluation framework of Proctor et al. 2011 [30], a mixed-methods implementation evaluation was undertaken to assess users’ views on the acceptability and appropriateness of CCv.1 (Study I).

### 2.1. Methods

Study I was a convergent mixed-methods study, utilizing a cross-sectional quantitative questionnaire and semi-structured interviews with a subset of participants [31]. The study received ethics approval from the University of Sydney on 23 May 2018 [Protocol No. 2018/003]. Canteen service users were eligible to participate if they were aged 12–25 years; impacted by their own or a familial cancer diagnosis; had registered to use CCv.1. An invitation to participate in the quantitative study was sent via email to a total of 1714 registered users of CCv.1. Potential participants were provided with an information sheet and informed consent was obtained from all participants involved in the study before completing the questionnaire online. Participants who completed the quantitative questionnaire indicated whether they were willing to participate in semi-structured interviews; these participants were invited to take part in an interview.

#### 2.1.1. Questionnaire

Participant demographics were collected at the beginning of the questionnaire, including participant age, gender identity, postcode, and cancer experience (Patient/Survivor, Offspring, Sibling, Bereaved Offspring, Bereaved Sibling). Using the taxonomy of implementation evaluation, the constructs of appropriateness and acceptability guided the development of the questionnaire [30]. Appropriateness assesses whether a program or service is compatible and relevant to the user. Acceptability considers users’ satisfaction with the program or service [30].

Questions about the appropriateness of CCv.1 measured both compatibility and relevance. Questions about compatibility covered participants’ most frequent method of access (computer, tablet, or smartphone); overall frequency of use, using closed-ended responses (1 = every day to 8 = I have only used the platform once); and frequency of use of individual components of CCv.1, such as discussion forums, stories, and blogs (1 = never to 5 = always). Those participants who only used CCv.1 once were subsequently asked reasons for not using the platform with a list of closed responses. Relevance was assessed by asking participants’ reasons for joining, using a list of closed-ended options, and if they had previously used other online spaces to talk to peers about cancer, using a yes/no response. Acceptability was measured using a series of 10 author-developed evaluative statements about the participant’s experience using CCv.1; for example, participants were asked how useful, interesting, and safe they found CCv.1. Responses were given using a 4-point Likert scale (1 = not at all, to 4 = completely).

#### 2.1.2. Interviews

A qualitative semi-structured interview was undertaken with a sub-set of participants who completed the quantitative questionnaire, to provide a more in-depth understanding of the appropriateness and acceptability of CCv.1. Interviews were conducted over the telephone by a trained researcher (MN), audio-recorded, and professionally transcribed verbatim.

#### 2.1.3. Data Analysis and Mixed-Methods Integration

Quantitative data were analyzed using descriptive statistics and frequency distributions. The Likert data were transformed into binary variables, where “mostly” and “completely” were combined, and “often” and “always” were combined. The transcribed qualitative interviews were analyzed using NVivo 12 Pro (QSR International Pty Ltd., Burlington, Massachusetts, USA) by a trained researcher (JC). The interviews were summarized under the topics of appropriateness and acceptability to develop a more comprehensive understanding of participants’ experience using CCv.1 [32]. Results from the quantitative and qualitative analysis were triangulated, and illustrative quotes from the interviews are presented to provide details and elaboration on participants’ questionnaire responses [33].

### 2.2. Results

Questionnaire responses were received from 135 participants with an 8% response rate. Most participants identified as female (83%) and the mean age was 18.6 years. Participants were from the following categories: patient/survivor (*n* = 35, 26%), offspring (*n* = 46, 34%), bereaved offspring (*n* = 49, 36%), sibling (*n* = 9, 7%), bereaved sibling (*n* = 2, 1%) with 4% (*n* = 4) from multiple categories. Of the 135 participants, 20 (15%) young people also participated in semi-structured interviews.

#### 2.2.1. Appropriateness

Participants used multiple devices to access the first version of CCv.1, including computer (*n* = 97, 72%), via their mobile phone (*n* = 74, 55%), and tablet (*n* = 21, 16%). A quarter of participants (*n* = 31, 25%) reported using Canteen Connect more than once a month, with only 4% (*n* = 5) using CCv.1 once a week or more. Almost 20% (*n* = 25) of participants had signed up for CCv.1 but had not used it since that time. The most common reason for not using CCv.1 was that participants could not speak with a counselor when they went in (*n* = 6, 24%), and the platform was difficult to use (*n* = 5, 20%). The most common reasons participants reported joining CCv.1 were to connect with other young people like me (*n* = 69, 56%), to chat with others (*n* = 57, 46%), and because it is easy to get help online (*n* = 57, 46%) (Figure 2).

There was a strong convergence between the quantitative and qualitative data. In the qualitative interviews, all participants reported that peer support and the ability to connect with “people like me” was the most important aspect of being part of CCv.1. Participants said they could speak with people their age or with the same cancer experience and they would understand what they were going through.

“I read stories about how other people dealt with [their cancer experience], ... what they found useful and helpful. [I] just talked to people who could relate. You know, I could talk to, and they would understand what I’m going through.”[Patient]

The qualitative interviews provided further details on motivations for using CCv.1, with users motivated by specific needs that sometimes changed over time. These changes included a family member’s death, a change in their cancer diagnosis, or a change in their coping.

“I wasn’t coping, and then I was reminded that my pop always used to tell me that if I wasn’t doing okay that I could go onto Canteen Online and then I thought of that and then I thought that I really need to talk to someone. Otherwise, I’m going to break down, and it’s going to be harder to get back up.”[Bereaved Offspring]

The most used feature of CCv.1 was the stories (*n* = 76, 62%) and the forums (*n* = 75, 61%). Fewer young people used the blogs (*n* = 57, 46%) and online counseling (*n* = 43, 35%). The qualitative interviews provided clarification on the benefits of young people sharing their own cancer experience in the stories and forums in CCv.1. Sharing their story was a way of validating a young person’s feelings about their cancer journey and was used as a coping strategy.

“[Posting in the OHC] was somewhat therapeutic and it allowed me to move forward with how I could help my sibling with cancer.”[Sibling]

Participants interviewed also found benefits from supporting other young people on CCv.1.

“I’ve always been this person who likes to make a difference, and when someone’s sad, I want to help.”[Patient]

#### 2.2.2. Acceptability

Most survey participants said they would recommend CCv.1 to other young people (*n* = 88, 81%) (Figure 3). Almost 2/3 of participants found the OHC helpful (*n* = 65, 59%) and interesting (*n* = 81, 72%). Over half the participants found CCv.1 easy to use (*n* = 62, 60%). Most participants felt safe using CCv.1 (*n* = 91, 92%) and could safely share their cancer experience (*n* = 83, 84%). Half of the participants felt they received support from the community (*n* = 47, 50%) and just under half (*n* = 46, 47%) felt connected with other people on CCv.1.

Participants provided practical ideas for improving the usability of CCv.1. Ideas included the need for a section on CCv.1 advertising Canteen’s upcoming events and a live chat or a private message section as a way of communicating with other young people in real-time.

“... with Facebook, you can have like a group chat, and you can see like who’s online... So like having a chat section for your community.”[Offspring]

Participants felt that a group chat would increase response time on CCv.1, leading to increased engagement.

“... it wasn’t like a live chat with others... you’ve kind of got to wait for the response and it tends to be quite time consuming...”[Sibling]

Participants liked having online counselors available to them but wished their hours of work could be extended.

“Sometimes I’m up [late] saying I want to talk to someone and then I realize that [Canteen Connect] is not available for me.”[Bereaved Offspring]

The qualitative interviews also provided further information on participants’ suggestions for improving engagement with the CCv.1. Participants suggested CCv.1 needed to be mobile responsive or have an app, as participants were more likely to access the OHC more frequently from their smartphone or if they were not required to log onto the platform each time they wanted to use CCv.1. Participants requested the ability to filter the forum discussions by topic or cancer experience (Patient/Survivor, Offspring, Sibling, Bereaved Offspring, Bereaved Sibling) to enable access to tailored content when using the platform.

“If my profile is linked to... my identity as a Canteen [client], ... then there could be subsections of the forums that are tailored to me.”[Bereaved Offspring]

Participants suggested there needed to be a constant refresh of new topics and posts to maintain engagement.

“I just noticed that I was reading the same stories over and over again... I found that if I go on like once or twice a week, they’ll be less chance of me reading the same story.”[Bereaved Offspring]

## 3. Study II: Needs Assessment and Idea Generation

The results from the implementation evaluation in Study I combined with rapid innovations in technology since CCv.1 was launched led to a decision by Canteen to develop a new, more technologically advanced version of CC. The development of CCv.2 was conducted utilizing a participatory design framework. PD is a methodological framework that promotes users’ participation in designing a technological solution [34]. To ensure end-user needs are met, PD guarantees users are involved with all stages of developing digital technology. Using iterative PD processes increases the relevance of the technology to users and the likelihood the technology will align with users’ needs [35,36]. Involving young people in participatory research and design has also been shown to positively benefit the young person; for example, by increasing social justice awareness, self-confidence, and a young person’s sense of connectedness [37].

The development of CCv.2 utilized the four phases of PD proposed by Clemensen and colleagues (2017) [35], covering (1) needs assessment; (2) idea generation; (3) testing and re-testing; (4) evaluation. The needs assessment and idea generation phases of PD were conducted through workshops with young people using an Appreciative Inquiry approach (Study II). The Appreciative Inquiry workshops aimed to add depth of understanding to suggestions from Study I for improving CCv.1, by exploring the characteristics of an ideal OHC and the technology and design features that would allow this ideal OHC to be achieved in CCv.2 [38].

### 3.1. Methods

Study II was a qualitative study of semi-structured workshops conducted using an Appreciative Inquiry approach. Appreciative Inquiry is a participatory approach that provides an opportunity for facilitating change and exploring innovative approaches to enhance services and organizations [35,39]. Appreciative Inquiry is a 4-step strengths-based approach in which participants: (1) “discover” the best elements of a service that address user’s needs; (2) “dream” about what the service could look like in the future and generate propositions for an ideal service; (3) explore ideas for the “design” of a service that would enable the ideal to be achieved, and (4) determine the “destiny” of the service by establishing “what will be” [39]. The workshops delivered in Study II covered the discover, dream, and design steps of the Appreciative Inquiry approach. The University of Sydney’s Human Research Ethics Committee on 23 May 2018 (Protocol No. 2018/003) approved the study.

Eligible participants were young adult representatives on the Youth Cancer Services and Canteen’s Youth Advisory Teams (YATs). These representative act as consumer advocates for young people impacted by cancer across Australia. Based on their lived experience of cancer, the YATs guide strategy, policy, and advocacy at Canteen through participation in research and consultation on service delivery. Informed consent was obtained from all participants involved in the study.

#### 3.1.1. Workshops

Two 90-min workshops were held, facilitated by author MN and a second Canteen staff member with experience in evaluation, participatory design, Appreciative Inquiry, and online community development. A topic guide was developed for the workshops that focused on generating ideas for an ideal OHC that would promote engagement and a sense of community from users. The workshop allowed for discussion to ‘discover’ young people’s experiences using online resources to connect with other young people affected by cancer; promote ‘dreaming’ about the ideal engaged and connected OHC using a creative ‘blue sky thinking’ activity; and to ‘design’ the ideal OHC by presenting ideas for OHC features and exploring convergent and divergent views.

#### 3.1.2. Data Analysis

The workshops were audio-recorded with participant consent and transcribed verbatim before being imported into NVivo 12 Pro (QSR International Pty Ltd.). A thematic analysis of the data, conducted by a trained researcher (KC), used a semantic, inductive, and constructionist approach to coding and analysis [40]. Transcripts were read multiple times to build familiarity with the data and code at the level of each new idea or concept. Codes were reviewed and organized into categories and then into themes, with a researcher focus on highlighting young people’s positive experiences with OHC in line with the strength-based Appreciative Inquiry approach. Categories were structured around the ‘discover’ and ‘design’ steps and then interpreted as themes to represent the ‘dream’ OHC proposed by participants. Initial themes were checked against the data, and then data was re-summarized into the three themes presented in this paper. Selected quotations from the consultation workshops are presented as illustrative examples of each theme.

### 3.2. Results

Twelve young adult leaders with a mix of cancer experiences (63% personal cancer diagnosis; 36% familial diagnosis), gender (55% male; 45% female), and ages (range 19–26, mean = 22.6 years) participated in the two workshops. Participants reported a range of previous experience using OHC, from never using an OHC to acting in a moderator role for other young people within an OHC. Three themes of peer connection, flexibility, and accessibility were identified, which represent the features and content imagined for an ideal OHC.

#### 3.2.1. The Ideal OHC Would Bring Together Young People with a Lived Cancer Experience Who Can Connect through Sharing and Understanding Each Other’s Stories

In the workshop’s discovery phase, young people identified that accessing OHC addressed a need for users to understand the lived experience of cancer by connecting with other young people with a similar or shared experience. Reading others’ stories provided information about a cancer experience that went beyond medical information-seeking. Participants described specifically searching for information or interactions with other young people with as close an experience to their own as possible (e.g., being of similar age or cancer type), so they could get an insider’s perspective on what their own experience might be like.

“When I was first diagnosed, I was trawling through every page on the internet I could find. Not particularly chatting to other people, but reading other people’s blogs about people who had the same cancer as I had... I was finding people that had similar stories... just leading up to treatment ‘[be]cause obviously I didn’t know what to expect...”[Survivor]

By seeking similar others through an OHC, young people were able to broaden their support networks; this was particularly important for young people in small, regional communities or with rare cancer diagnoses.

“I think for people who want to find others who’ve had similar experiences, it’s kind of rare for cancer patients to find that in the real world, so a dedicated online space is probably ideal for that.”[Survivor]

In the design phase, young people recommended several features of an OHC that would allow users to easily connect with similar others, including filtering community discussions on key demographics and prioritizing these discussions when viewing the OHC; creating sub-communities of users that could be followed, and enabling hashtags so people could search by topics of interest. These features would help build connectivity to others and increase relevance and engagement, while still allowing young people to access the full range of information and stories if needed.

“If you could get different people to write stories, and they have an ability to filter that, by like, age, diagnosis, gender, or whatever you want—to be able to kind of find something that fits you. So, you do kind of get that more connection, rather than just, like, someone else with a different cancer...”[Survivor]

Privacy was an important part of the ideal OHC and building connections between young people. In the discovery phase, young people described their ideal OHC as being open only to other young people who experienced a personal or familial cancer diagnosis. The participants preferred that other people interested in a youth cancer experience, such as educators, health professionals, or researchers, use public information or resources rather than having access to an active OHC. Participants described that this eligibility requirement for an OHC would enable them to trust that they were giving and receiving support from people who understood their reality. In turn, this would enable them to be more vulnerable in sharing their stories and increase engagement on the OHC.

“I think the information should probably be accessible for anyone, but I don’t know whether the interactive side should be... the discussion stuff and the chats and all of that should be private.”[Survivor]

In the design phase, participants suggested that community privacy be protected by having people complete a registration process before being able to access the community. Including simple questions about cancer experience in the registration process, such as cancer diagnosis or treatment type, was intended to signpost that the community was for people with a lived experience of cancer. Participants recognized the need to balance privacy concerns with support needs. For example, participants prioritized young people seeking urgent mental health support having immediate access to the community and counselors over adding steps to confirm people registering for the community had a lived cancer experience.

Once part of the community, young people suggested having the option to choose (and change) a username and the opportunity to post under this username or anonymously, depending on the topic. Participants wanted clear information about how and where their personal data would be used; this was perceived as a crucial requirement for giving young people the security to engage in personal discussions and provide support.

#### 3.2.2. The Ideal OHC Would Allow Young People the Flexibility to Access Different Types of Support to Suit Their Current Needs in a Supported Environment

In the workshop’s discovery phase, young people described having different types of support needs at different times throughout their cancer experience; for example, seeking medical information at diagnosis, connecting with others during treatment, or giving support back to others during times of personal or family wellness. Participants identified a strength of OHC as being able to offer these various kinds of support in a single place. This capability was particularly helpful for young people living in areas that have fewer in-person support options.

“So for me, I get that initial point, like cancer’s really, really important, and that face-to-face connection was really important for me, but then, as I went along, you know, I didn’t need that anymore, but then I was like, ‘Oh, you know, online counseling is great’ or ‘phone counseling is great.’ So if you know where everything is available up front then you can kind of choose what you want, when you need it.”[Survivor]

In the design phase, participants suggested addressing varying needs by offering support in a range of formats, such as chat, phone, or video counseling. They further recommended having instant or live connections with other users or counselors, so young people could access support without having to wait for replies or an appointment.

“I think [live chat] would make a really cool addition. It’s instant. If there is someone online, you can talk to them straight away. Another user, or even a counselor to have someone to talk to.”[Bereaved Offspring]

While participants extolled the benefits of connecting with others like them, they also recognized that some users of OHC would have needs that required additional or professional training to address. For example, participants wanted to ensure that young people experiencing mental ill-health received specialist support. Participants suggested using screening tools or algorithms to automatically monitor the community for signs of distress and direct users to personalized forms of support. This support could come from staff moderating the community and/or young people who were trained to act as peer moderators. Peer moderators were regarded as providing a benefit by being able to respond outside business hours; provide support on the merit of their lived experience; increase real-time community activity and engagement; empower young people as leaders in the community when they were ready to give back to others. Participants acknowledged the ideal OHC would balance young people and professionals’ roles so that peer moderators were not required to manage interactions beyond their expertise.

“I go on quite often. I will check to make sure that everyone’s post is being replied to. If I can’t reply at the time, I’ll at least acknowledge that it’s there... if it’s a quiet week, I go on two times to check that no one is being left behind.”[Bereaved Offspring]

Participants also emphasized the importance of professional input into information about cancer on the OHC to ensure veracity. Participants suggested the need for a frequently-asked-questions section with scientifically validated information in age-appropriate language.

#### 3.2.3. The Ideal OHC Would Be Easy to Access and Become Part of Young People’s Online Activity Routine

The workshops’ discovery phase highlighted that young people accessed support from online resources and OHC because of their convenience. The online format enabled young people to extend their access to resources beyond the local community and removed requirements to locate and store physical resources (e.g., pamphlets, information books). Young people with a personal cancer diagnosis particularly described the convenience of using OHC when they did not feel comfortable or able to go outside due to the physical or psychological effects of treatment.

“I think cancer’s a very confronting issue to talk about. And so, going online tends to give you a bit of distance. It can be helpful unless you’re really comfortable, you can go and do it in your bedroom. A space in which, I suppose, in which you’re safe.”[Survivor]

Young people were more likely to engage with OHC when they became part of their “regular lives” due to ease of access and community activity levels. In the design phase, young people recommended simplifying the registration and sign-in process as much as possible to allow quick access to the OHC. Participants specifically recommended using single sign-on technology or integration with other apps to allow ‘one button’ access to the OHC, increasing the likelihood that young people would regularly check discussions and add to the activity. Ultimately, participants wanted the Canteen OHC to be accessible via an app.

“I really think there should be an app. I think the fact that everyone has a phone—I know if it was on an app I would use it... I just feel like the app brings so much. The notifications straight to your phone. That’s how you’re going to get the interaction from people.”[Survivor]

## 4. Study III: Implementation Evaluation of Canteen Connect Version 2

The OHC developers used the findings from the implementation evaluation of CCv.1 (Study I) and the needs assessment and idea generation (Study II) to develop an initial prototype of the second version of CCv.2. Changes from CCv.1 aimed to improve user experience, making it easier to navigate and access the OHC. These changes included the ability to search specific topics in the discussion forums, the site preferentially showing users topics related to their cancer experience, the ability to direct message other users, and the site becoming mobile responsive. Additional changes to CCv.2 included upgrading the functionally so users could more easily access online counselors and having Canteen events more readily displayed on the platform, with users showing events related to their cancer experience and location. The story and blog features were removed. Peer moderators were also introduced to CCv.2 as part of a dedicated leadership team, aiming to increase engagement by sharing their cancer experiences with the community and helping to lead discussions. A feature was also introduced giving users the option to make their discussion posts publicly available or to keep them private within the CCv.2 community, giving users more control over the audience for their discussions.

In the testing and re-testing phase of PD, rapid prototyping and testing with key stakeholders was completed using an agile methodology to refine the final design of CCv.2. A critical process in a technical healthcare solution’s PD is evaluating its delivery in a clinical or real-world setting [35]. Once CCv.2 had been launched, we undertook the evaluation phase of the PD process. Using an implementation evaluation framework [30], we assessed the appropriateness, acceptability, and preliminary effectiveness of CCv.2 (Study III).

### 4.1. Methods

The cross-sectional quantitative study used the implementation evaluation framework of Procter et al., 2011 [30] to assess the appropriateness, acceptability, and effectiveness of CCv.2. The study received ethics approval from the University of Sydney on 2 July 2020 [2020/361]. Eligible participants were Canteen service users who were registered for the second version of CCv.2. A total of 1635 young people were invited via email to participate in the study. The email included a link to the study information sheet, and the questionnaire was hosted on a secure online survey management platform. Informed consent was obtained from all participants involved in the study.

#### 4.1.1. Quantitative Questionnaire

The questionnaire included information on demographics, appropriateness, acceptability, and effectiveness. Participant demographics covered participant age, gender identity, postcode, and cancer experience (Patient/Survivor, Offspring, Sibling, Bereaved Offspring or Bereaved Sibling). Appropriateness was assessed as per Study I, with questions about participants’ frequency of use; use of core features; reasons for joining; reasons for discontinuing use. Participants were also asked how useful they found CCv.2 for helping them to (1) feel connected; (2) access a counselor; and (3) get information on their cancer experience; responses were provided on a 4-point Likert-scale (1 = not at all useful to 4 = very useful) with an additional N/A response.

Key indicators of CCv.2′s acceptability was participant satisfaction, measured on a 1–10 scale (1 = not at all satisfied to 10 = extremely satisfied), and if they would recommend it to other people like themselves (Yes/No). Acceptability was assessed as per Study I, with evaluative questions about participant experience measured on a 5-point Likert scale (1 = strongly disagree to 5 = strongly agree). Effectiveness was measured by asking whether participants felt CCv.2 helped with their feelings of sadness, worry, anxiousness, lack of connection, and not feeling understood [23]. Participants were initially asked whether they had issues in these areas in the past four weeks. Participants who responded affirmatively were then asked to rate the perceived extent that being on CCv.2 had impacted these feelings, using a 5-point Likert scale (1 = made it much worse to 5 = made it much better).

#### 4.1.2. Analysis

Quantitative data were analyzed using descriptive statistics and frequency distributions. The Likert data were transformed into binary variables, where “mostly” and “completely” were combined, hereafter known as “completely”, and “agree” and “strongly agree” were combined, hereafter known as “agree”. Responses from the open-ended questions were reviewed and categorized.

### 4.2. Results

A total of 120 participants completed the implementation evaluation. Most participants identified as female (*n* = 88, 73%), and the average age was 19.3 years (±3.6 years). A total of 34% of participants (*n* = 39) were from rural or remote areas. Participants were from the following categories: patient/survivor (*n* = 40, 39%), offspring (*n* = 30, 29%), bereaved offspring (*n* = 25, 24%), sibling (6, 5%), bereaved sibling (*n* = 4, 4%).

#### 4.2.1. Appropriateness

Over half of the participants used the second version of CCv.2 for longer than six months (*n* = 66, 55%), and 19% (*n* = 23) had been using CCv.2 for less than three months. A total of 52% of participants (*n* = 60) reported using CCv.2 more than once a month, with 29% (*n* = 33) using CCv.2 once a week or more. Only 10% of participants (*n* = 11) reported using CCv.2 once. Of those who said they only used CCv.2 once, the most common reason given for this was that they logged on as they were curious (*n* = 6, 45%), rather than an issue with the content or features. Almost three-quarters of participants (*n* = 71, 71%) selected more than one reason for joining CCv.2. The most common reason for joining was to connect with young people like themselves (74%) (Figure 4).

Young people selected which of the six core CCv.2 features they used, with the option to select multiple features. The most used CCv.2 features were the discussion forums (*n* = 79, 66%) and the events section (*n* = 53, 44%) with 85% (*n* = 93) reporting they used more than one feature on Canteen Connect. The feature to speak with an online counselor was used by 33% of participants (*n* = 39). The ability to use private messaging was a new feature introduced to CCv.2, with 26% (*n* = 31) of participants reporting they used the private messaging feature. A total of 74% (*n* = 70) of participants found CCv.2 useful for connecting with young people like themselves; 84% (*n* = 53) said it was useful for chatting with a counselor, and 62% (*n* = 58) found it useful for finding information about their cancer experience.

There was strong convergence between the quantitative results and the responses from the open-ended questions. From the open-ended questions, the most liked feature of CCv.2 was the ability to connect with young people like themselves. Participants also thought CCv.2was accessible and user-friendly, making it easy to navigate and make these connections with other young people with a similar cancer experience.

“I like that it is an easy platform to use for young people to connect with others who have been through a similar experience.”[Bereaved Offspring]

Participants also commented the events section was useful and it was easy to access a counselor on CCv.2.

“The access to counselors is really easy and approachable at any time you need support, and it’s super comforting and helpful to know that they’re there specifically to talk to you about your cancer struggles.”[Bereaved Offspring]

#### 4.2.2. Acceptability

There was a mean satisfaction rating of 7.43 (±2.3), with 71% of participants (*n* = 77) rating CCv.2 as seven or higher, where 10 equals extremely satisfied. Over 90% of participants said they would recommend CCv.2 to other young people like themselves (*n* = 99, 91%). Most participants felt safe using CCv.2 (*n* = 102, 92%), and 72% (*n* = 80) felt they received support from the CCv.2 community (Figure 5). Almost three-quarters of participants agreed that CCv.2 was helpful (*n* = 87, 77%), interesting (*n* = 90, 77%) and easy to use (*n* = 79, 71%).

Results from the open-ended questions highlighted several recommendations for further improvements to CCv.2. Consistent with recommendations from the evaluation of CCv.1, one of the most common recommended changes was to develop an app to make it easier to access CCv.2 without the need for a computer.

“Make an app!!! I would use Canteen Connect so much more if there were an app as it is so much easier to use.”[Bereaved Offspring]

Participants recommended that Canteen continue to work on the interface and navigation of CCv.2 to improve the user experience further.

Some users also felt that there was not always great engagement when they posted in the discussion forums.

“[I] wish there was a bit more advice/help suggested for me via discussions as whenever I posted something it hardly got read, or people seemed to disregard my post/not reply.”[Offspring]

Participants suggested that there could be moderated live chats in the discussion forums or separate “chat rooms” where young people with a similar cancer experience could connect. Participants also suggested the need for content in the forums’ discussions to be continually updated and refreshed.

“I feel like a lot of the posts relevant to me are quite old so I don’t comment because it feels like no one would see it.”[Sibling]

#### 4.2.3. Effectiveness

Most young people reported experiencing difficult feelings due to their cancer experience in the past month, with 75% experiencing sadness (*n* = 81), 60% experiencing worry (*n* = 65), and 64% (*n* = 70) experiencing anxiety. Almost half of the participants (*n* = 53, 47%) reported experiencing issues with all three feelings, and 13% (*n* = 15) reported experiencing issues with two feelings. Participants experiencing issues were asked how using CCv.2 had impacted their feelings. Over 60% felt that using CCv.2 positively impacted their feelings; fewer than 4% of young people felt that using CCv.2 negatively impacted their feelings of sadness, worry, and anxiousness (Figure 6).

Responses from the open-ended questions emphasized how CCv.2 helped participants with their cancer experience.

“I really love and appreciate the idea behind Canteen Connect and I also really valued speaking to [online counselor] once when I was having a low point during my cancer journey.”[Bereaved Offspring]

For participants who reported they felt misunderstood (76%, *n* = 83) or did not feel connected with other people (47%, *n* = 51) before using CCv.2, over 70% reported that CCv.2 helped with these feelings. The open-ended responses highlighted how CCv.2 helped improve their feelings of connection.

“It makes you realize you are not alone, and other people share the same fears and worries you do.”[Offspring]

## 5. Discussion

Canteen Connect is the first OHC designed to support young people impacted by their own or a familial cancer diagnosis. A PD approach was used to develop the OHC to optimize its appropriateness, acceptability, and effectiveness, ensuring that the latest version met the needs of young people impacted by cancer. Results from the needs assessment highlighted the need for participants of an OHC to connect through sharing and understanding each other’s stories; to access different types of support to suit their current needs in a safe, structured environment; and to have an OHC that is easy to access and be flexibly integrated into a young person’s lifestyle. The idea generation and test-retest phases of the PD process highlighted the importance of finding novel ways to maintain engagement within the community, such as the introduction of peer moderators. Results from the implementation evaluation showed that the latest version appropriately addressed young people’s needs for peer connection and emotional support. This version was also seen as (1) acceptable to users, bolstered by design and features that allowed young people to flexibly engage with the platform in a way that suited their current needs, and (2) an effective technology for young people to receive emotional support, with positive impacts on young people’s feelings of sadness, worry, and anxiety.


*Appropriateness for addressing peer connection needs*


Across all three studies, young people consistently highlighted that they would or did join an OHC to receive peer support and connection from other young people. Connection with others can help to reduce young people’s social isolation by normalizing and validating their experiences [13]. Young people specifically highlighted seeking connection from others with as close an experience to their own as possible, such as those with a similar age or cancer type. This is consistent with previous research on other types of OHC [14]. Access to similar peers can be challenging in a country like Australia, which has a dispersed population and limited mental and physical health services in rural and remote areas [14]. Providing peer support via an OHC removes geographic barriers and can increase the likelihood of young people finding support from someone with a similar experience [14].

Results from the implementation evaluation of CCv.1 found that while participants who used the OHC felt safe and that they could share their cancer experience, only half of the participants felt connected or supported by other young people in the community. The introduction of a PD process in the development of CCv.2 aimed to increase the appropriateness of CC in addressing young people’s needs for peer support. Across the three studies, young people made several suggestions for features that could be implemented to increase user engagement and peer connection. These included more frequent updating of topics and posts; group chats for specific cancer situations; private messaging between users; volunteer moderation from young people affected by cancer. These user-generated changes were theoretically predicted to increase engagement and facilitate connection. For example, frequent content updates influence whether individuals join, post to, and ongoingly use, an OHC [41], with sustained OHC activity building a sense of community and belonging [13]. Further, using a combination of professional and volunteer leadership has been hypothesized to facilitate a sustainable community in OHC [16]. Following the changes made for CCv.2, around three-quarters of evaluation participants said they found the OHC useful for connecting with other young people and that they received support from the community. This provides evidence of the utility of the PD process for ensuring that an OHC is appropriate for addressing peer support needs in young people affected by their own or a familial cancer diagnosis.


*Appropriateness for addressing flexible emotional support needs*


The psychological impact of cancer for young people is well-noted, with more young people reporting high levels of psychological distress than their non-cancer-affected peers [3,4,5,7]. Young people seeking support to manage difficult emotions may turn to friends, peers, and professionals; these types of support can all be made available on an OHC. The PD process suggested several key changes to CC that could improve opportunities for young people to receive emotional support from peers and professionals via the OHC. Peer interactions were supported in CCv.2 with the introduction of direct messaging capabilities between users. Increasing opportunities for peer connection has the potential to improve access to emotional support and to improve coping [16]. For CCv.2, platform functionality was upgraded so users could more easily access online counselors, with results from the Study III implementation evaluation providing evidence of this improved accessibility. Although online support groups cannot always replace offline relationships [16], the availability of online counseling as part of the OHC can provide an essential link between online and offline support. The results from the Study III implementation evaluation showed an increase from Study I to Study III in the proportion of participants saying they received emotional support from the community.

By providing multiple types of support, the benefit of an OHC is that it can meet diverse needs across users, as well as meet changing needs within individual users. Participants across Studies I and II reported varying support needs, including passive information-seeking, seeking support through interactive discussions, seeking support from individual counseling, and providing support to others in a ‘mentor’ role. Participants further described wanting an OHC that would allow them the flexibility to access different types of support as their current needs changed. The concept of multiple, changing needs was supported by findings from Study III, with most participants reporting they used multiple features of CCv.2. Providing a range of individual support options and types of interaction within these options (e.g., moving from user to peer moderator) allows CCv.2 to meet the variable needs of users, rather than adopting a ‘one size fits all’ approach [41]. This is likely to increase community engagement, as previous research suggests that user activity is positively influenced if users can both receive and provide support within the community [42,43].


*Acceptability and sustained community engagement*


To ensure an OHC is acceptable to users, the functionality of the platform must support a level of activity that can build and sustain community engagement [44]; this fosters the sense of community needed to achieve the social benefits of an OHC [16,45]. It is common for users of an OHC to be transitory, with only 10% of users likely to actively contribute to discussions [46]; this means that an OHC must be acceptable to many registered users for sufficient community activity and engagement to occur. There are also significant resources required to ensure a critical mass of users within an OHC to maintain a sustainable community [16]. Feedback on CCv.1 from Study I indicated limited platform acceptability from young people. In part, this was attributed to low rates of community activity, such as infrequent posting or interactions between young people, caused by accessibility issues with CCv.1 (e.g., limited search functionality, lack of direct messaging).

Participants in Study II highlighted areas that could be changed to improve ongoing community activity and engagement and therefore the OHC’s acceptability. Specifically, participants described that they would be more likely to engage with an OHC that was highly trafficked due to ease of access, ease of interpersonal connection, and integration with young people’s regular online activities. Following the PD process, these ideas were incorporated into the development, testing, and re-testing stages for CCv.2. A key feature upgrade from CCv.1 to CCv.2 was mobile responsiveness, enabling users to access CCv.2 at any time and location on their smart phones. Ease of connection was improved by optimizing the search function in the discussion forums, allowing users to more readily access discussions related to their own cancer experience, as well as priority display and filtering of community discussions based on relevant demographics for the user. Results from Study III showed CCv.2 was acceptable to users, as evidenced by the high level of satisfaction reported. A greater percentage of users found CCv.2 easier to use than CCv.1. There was also a greater percentage of users using CCv.2 more than once a week compared with CCv.1. These increases in acceptability highlighted the importance of the PD process for generating novel ways to maintain engagement within the community. This process is ongoing; users have recommended the development of a CC app to further increase accessibility, acceptability, and engagement.


*Effectiveness at improving psychological well-being*


Previous research on OHC in general populations of young people has found that discussions encompassing emotional expression, emotional support, and practical information [47] could lead to improved health-related or psychosocial outcomes, such as coping [23]. However, there is mixed evidence in the literature to show whether an OHC can have positive psychosocial impacts for people impacted by cancer [16]. Study III provided preliminary evidence for the effectiveness of CCv.2, with most participants experiencing worry, sadness, or anxiety reporting that these feelings had improved after using CCv.2. It may be that increasing accessibility to online counseling as part of CC v.2 has improved psychosocial outcomes for users, along with the combined ability to receive emotional support from peers. Research also suggests that online peer support should be an adjunct to offline support and wellbeing may be better for high-risk users who seek a combination of online and offline psychosocial support [48,49].

Very few participants in Study III indicated that using CCv.2 had made their feelings of worry, sadness, or anxiety worse. There have been concerns that using OHC could put users at risk for worsening of symptoms, exposure to misinformation, privacy issues, or conflictual peer interactions [16]. Users’ exposure to other users’ negative experiences and stories may increase the risk of negative psychosocial consequences for peers in the community [50]. The use of PD in the redevelopment of CCv.1 enabled features to be put into place to promote psychosocial outcomes and reduce potential risks; for example, improved access to online counselors, and moderation of discussion forums by trained psychosocial clinicians.

### Limitations

The participants in this study represent a small proportion of users of CCv.1 and CCv.2. Although response rates of 8% appear to be similar across Study I and Study III, these are slightly lower than the 13–15% seen in similar studies [23,24]. It is possible that only young people who were most engaged with CCv.1 and CCv.2 or with Canteen chose to participate in these self-report studies and that their perspectives do not represent all users, particularly those who are dissatisfied or disengaged. Future studies evaluating the implementation and outcomes of OHCs could consider targeting less engaged users, or using objective indicators, such as login and activity rates, to measure engagement with CCv.2. The current study did not collect data on participants’ cancer diagnosis or time since cancer diagnosis. As factors such as cancer diagnosis and stage of the disease could influence the psychosocial impact of an OHC [16], future studies could collect additional demographic data to further understand whether the individual needs of young people are being met. Most respondents in Studies I and III identified as female. Although this is representative of users registered on CCv.1 and CCv.2, it may indicate that this OHC is more likely to be accessed by specific groups of AYAs within the broader target population. Future work could examine routinely collected user data to test this hypothesis. Further, future studies could consider the use of purposive sampling based on demographics or oversampling from minority user groups to ensure that all users’ perspectives are represented in future platform development. This study provides preliminary evidence of the effectiveness of CCv.2, though it remains unknown whether being part of CCv.2 can provide sustained psychosocial benefits. Future work is needed to assess whether young people impacted by cancer who use CCv.2 for peer connection and support have improvements in long-term psychosocial outcomes.

## 6. Conclusions

Young people living with and beyond their own or family member’s cancer need safe, secure, and accessible ways to connect with their peers and access information, peer, and professional support. The experiences reported by users of CC present important evidence about the need for diversified approaches to AYA mental health support to complement traditional face-to-face modalities of therapeutic and/or psychosocial support. CCv.2 provides this forum for young people, with evidence for its acceptability and appropriateness, and preliminary evidence for its effectiveness.

The story of CC highlights the advantage of using a PD process to develop an OHC that promotes engagement from users by finding novel ways to facilitate connection and maintain engagement within the community. Utilizing the PD steps of needs assessment, idea generation, testing, re-testing, and implementation evaluation meant that changes to design features from CCv.1 to CCv.2 were user-generated and therefore more likely to meet users’ needs. The introduction of design features, such as easy access to online counselors; optimization of the search function in the discussion forums allowing users to access discussions that relate to their own cancer experience; the ability to filter community discussions on key demographics; having the capacity to prioritize viewing of targeted discussions for users; and the capacity to private message other users allowed young people to flexibly engage with the platform in a way that suited their individual needs. An OHC that can target a range of support needs and formats also ensures ongoing engagement and connection within the community.

The findings of this evaluation demonstrate that ongoing resourcing and investment in an OHC is needed to drive sustained use and engagement within the community. Future studies could collect detailed information on the ongoing monetary and resource costs required for an OHC, as well as the cost-effectiveness of an OHC in supporting the psychosocial needs of AYAs impacted by cancer. As there is little research assessing the health economics of maintaining a sustainable OHC, this would provide important information to assist other organizations in deciding whether to implement similar OHC.

Since the completion of Study III, ongoing changes to CCv.2 suggested by the implementation evaluation have been adaptively implemented to meet users’ needs. The CC app has also been recently launched to facilitate real-time engagement among users on CCv.2. There are plans to introduce group chats within CCv.2 and work will continue with stakeholders to identify novel ways to improve and sustain engagement within the CC community. Finally, a longitudinal study is underway assessing the longer-term impact of using CCv.2 on a young person’s distress, loneliness, peer support, and sense of community. This body of work serves young people impacted by cancer by ensuring that effective technology is in place to promote thriving and positive well-being through receiving and giving emotional support.

## Figures and Tables

**Figure 1 cancers-14-00050-f001:**
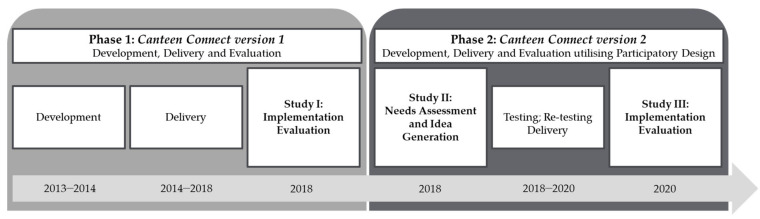
Components of the development, delivery, and evaluation of Canteen Connect.

**Figure 2 cancers-14-00050-f002:**
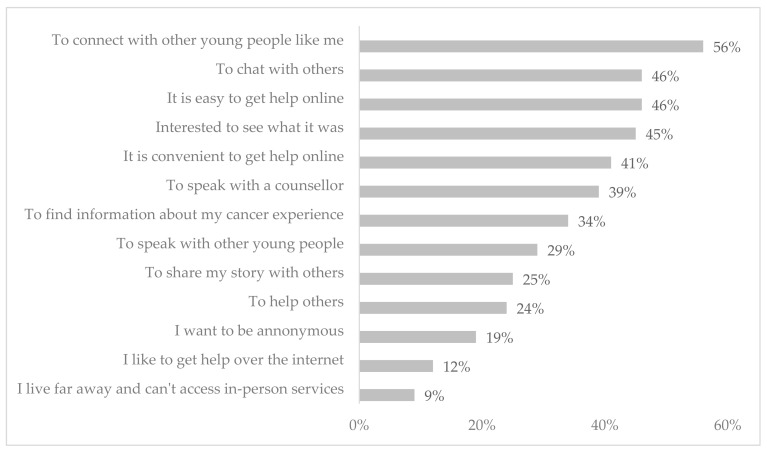
Reasons for joining Canteen Connect version 1.

**Figure 3 cancers-14-00050-f003:**
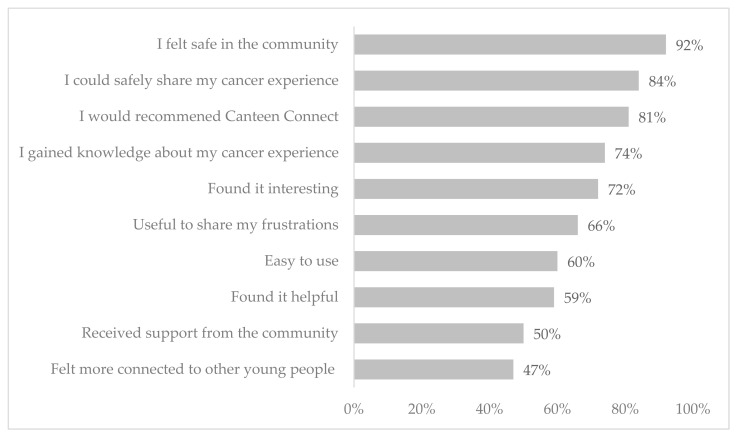
Acceptability of Canteen Connect version 1.

**Figure 4 cancers-14-00050-f004:**
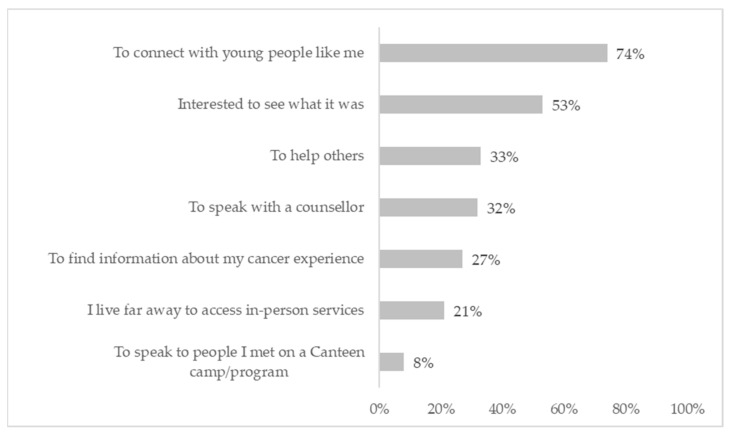
Reasons for joining Canteen Connect version 2.

**Figure 5 cancers-14-00050-f005:**
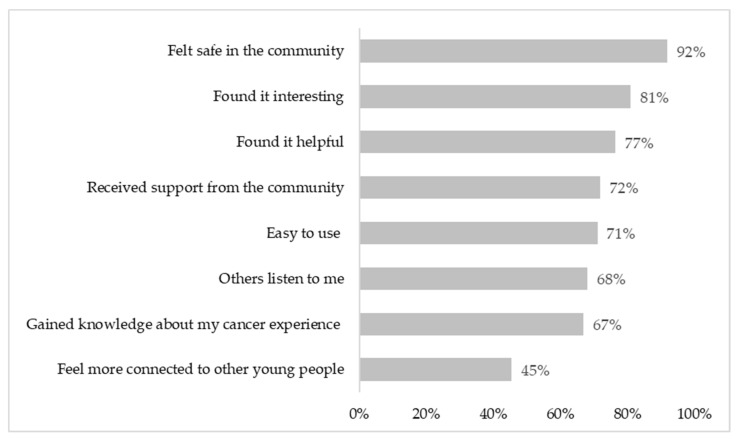
Acceptability of Canteen Connect version 2.

**Figure 6 cancers-14-00050-f006:**
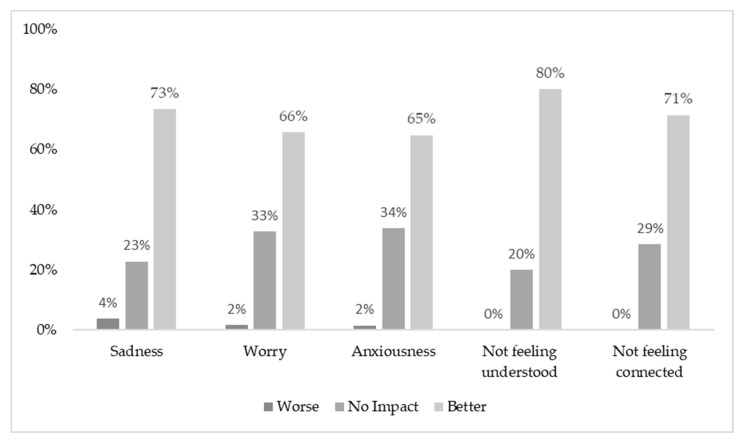
Impact of Canteen Connect version 2.

## Data Availability

The data presented in this study are available on request from the corresponding author. The data are not publicly available due to ensure privacy.

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
