# Peer review of "Development and Evaluation of the Canteen Connect Online Health Community: Using a Participatory Design Approach in Meeting the Needs of Young People Impacted by Cancer"

_cancers, 2021, doi:10.3390/cancers14010050_

Round 1

Reviewer 1 Report

This is a very well-written paper on the development, delivery, and evaluation of Canteen Connect online health community (OHC) project that used a participatory design approach to meet the needs of young people impacted by cancer in Australia. On line #101, the authors state that in 2014, Canteen launched an OHC for young people impacted by personal or familial cancer. However, the Patterson, McDonald, and Orchard study (Reference #26 in the current paper) was already published in 2014, and they used the term “CanTeen” instead of “Canteen”. The authors of this paper may want to clarify these minor inconsistencies here.

My bigger concern is about the total lack of information about cancer types. Cancers affecting adolescents and young adults are expected to be different from those affecting the parents of young people. Risk of some cancers such as breast and colorectal cancers increases with age and cancers such as cervical and prostate are specific to gender. Line #125 in this paper mentions that demographic characteristics such as age, gender, postcode, cancer experience etc. were collected for this study. To get a better sense of the individualized experiences, the authors could provide a descriptive summary table with more details on the participants including their own or their siblings’ or parents’ experience with cancer type and mean time since diagnoses, their employment and education status, and other relevant variables. Also, a recent paper by McCann, McMillan, and Pugh, provides a systematic review of digital interventions to support adolescents and young adults with cancer [JMIR Cancer 2019;5(2):e12071], which can be used by the authors to examine current literature in this area and provide better context for their study. Additionally, the paper by Kaal, Husson  et al. (Reference #20 cited in this paper) mention that of the six adolescents and young adults (AYA) communities/websites currently available with different features to exchange informational, emotional, and social support (described in their  Table 1), none provides a secure environment. The information from this reference could be used by the authors to highlight if this is a strong point of the Canteen Connect OHC tool. Finally, there is no discussion in the paper about the resource requirement of the Canteen Connect OHC project that could be an important first step for a future cost-effectiveness study. The authors could mention this as a current evidence gap.

I am providing other minor comments below for the authors’ review.

The introduction could be enriched with some statistics on the higher burden of depression on adolescents and young adults in Australia who are impacted by cancer. Also, a previously published paper on Canteen Connect by Patterson, McDonald, and Orchard (Reference #26 in the current paper) should be introduced at the beginning of this study to provide the right context.

The methods, analysis, and results sections of Study I and Study III include some repetitive sentences. The authors may want to paraphrase or shorten some of these sentences in Study III that were presented  before.  

Line #38 (Abstract): Add a “to” between “message” and “others” in “…such as the ability to direct message other users…”

Line #101: Please see my previous comment. Also, Figure 1 shows that CCv.1was developed in 2013 and not 2014.

Line #129: Appropriateness is defined as whether a program or service is compatible and relevant to the user. The type of cancer type as I discussed earlier could be very relevant here.

Lines #248-249: It is stated that “participants requested the ability to filter the forum discussion by topic or category to tailor  their experience on the platform.” Again, information on the type of cancer could provide a useful context for this statement.

Line #325: For Study II, the participants were predominantly male (55% male; 45% female) whereas those for Study I and Study III were predominantly female. Although results from the implementation evaluation showed that the latest version appropriately addressed young  people’s needs for peer connection and emotional support (lines #644-646), it is not clear if this non-uniformity in gender distribution affected the needs assessment and idea generation in any specific way. In particular, male participants would have knowledge of female cancers only indirectly through siblings or parents who may be affected by these cancers. I understand that there is limited choice in recruitment of workshop participants; however, the authors may want to comment if this a problematic issue.

Reviewer 2 Report

Thank you for the opportunity to engage in this review. I enjoyed reading the manuscript and learning of this work. It is so important to tailor programming to the audience/population and to engage members from that group in designing any intervention. 

I think the paper is well written. It is clear what has been done and how the three studies build on one another. The only comment I might add relates to rigor around who was engaged in the coding and analysis of the qualitative transcripts. It is important to reflect whether the work was done by one or more of the investigators (who had what roles regarding coding and analysis).

The results are well presented and the discussion is appropriate. You might had made mention about whether your response rates on the questionnaires were a concern to you or not, given they were so low (8%). 
